# Emergent Inabilities? Inverse Scaling Over the Course of Pretraining

**James A. Michaelov** and **Benjamin K. Bergen**
Department of Cognitive Science, University of California San Diego
{j1michae,bkbergen}@ucsd.edu

## Abstract

Does inverse scaling only occur as a function of model size, or can it also occur over the course of training? We carry out an exploratory study investigating whether the performance of language models on specific tasks can decrease (while general performance remains high) during training on the language modeling task. We find 8 tasks on which Pythia 12B (Biderman et al., 2023) shows decreased performance over the course of training. Five of these tasks (TRUTHFULQA-MC1, TRUTHFULQA-MC2, HINDSIGHT NEGLECT, MEMO TRAP, and PATTERN MATCH SUPPRESSION) additionally show a consistent relationship whereby larger language models show a greater decrease in performance the more they are trained, despite showing standard (positive) scaling overall. This highlights the importance of testing performance at all relevant benchmarks any time models are trained on additional data, even if their overall performance improves.

## 1 Introduction

For language models, bigger is usually better. Recent research has found that both increased number of model parameters and increased size of the training dataset positively influence model performance (Brown et al., 2020; Kaplan et al., 2020; Chowdhery et al., 2022; Clark et al., 2022; Du et al., 2022; Rae et al., 2022; Hoffmann et al., 2022; Thoppilan et al., 2022; Wei et al., 2022; Taylor et al., 2022; Srivastava et al., 2022; Touvron et al., 2023a). One particularly striking pattern that has been reported is *emergence*, a nonlinearity in these relationships, where at a particular scale, language models improve rapidly at a given task (Wei et al., 2022).

However, while increased scale usually leads to improved performance, on certain tasks it correlates with decreased performance. This is known as *inverse scaling* (Lin et al., 2022). An example of a task on which inverse scaling is observed is the TruthfulQA benchmark, where larger language models are more likely to predict popular misconceptions over statements of fact (Lin et al., 2022). More recently, additional tasks that reportedly show such an effect have been identified as part of the Inverse Scaling Prize (McKenzie et al., 2023b), as well as by other researchers (Jang et al., 2023; Michaelov and Bergen, 2023).

Inverse scaling is a serious concern for several reasons. At a high level, inverse scaling may indicate 'outer misalignment' (Perez et al., 2022) between the model training approach and the purposes to which they are applied. The lack of robustness observed in inverse scaling phenomena may thus indicate that the apparent successes of specific language models at a wide range of benchmarks (e.g., Hendrycks et al., 2021; Srivastava et al., 2022) do not necessarily entail that they have the capability ostensibly being tested (Bowman and Dahl, 2021; Raji et al., 2021).

The existence of inverse scaling is also concerning because of the possibility of other as yet unidentified tasks where performance similarly scales inversely with model size. Models that perform well on a variety of tasks may well present deteriorating performance in unseen tasks with scale, even as performance at established benchmarks increases. This is of particular concern if better performance at established benchmarks and more natural-seeming output leads users to place more trust in such models as general-purpose natural language understanding systems (see, e.g., Bender et al., 2021, for general discussion of such risks).

Finally, inverse scaling is also of concern because it is often unpredictable. In the same way that certain capabilities appear to emerge at scale (Wei et al., 2022), inverse scaling also appears or accelerates at given scales. For example, as McKenzie et al. (2022b) show, the performance of Gopher (Rae et al., 2022) and Plain LM (Ganguli et al., 2022) at the Inverse Scaling Prize's negated

question-answering task (NEQA) appears to be stable or even increasing as model size increases, only dropping as model size increases to around 7 billion parameters and beyond (McKenzie et al., 2022b). Thus, inverse scaling may occur not just for unidentified tasks, but also for well-established tasks: a model architecture that performs well at a benchmark at a small scale may suddenly perform surprisingly worse as scale increases–it is not safe to assume that performance will continue to improve or even remain stable.

While previous work has focused on inverse scaling based on the number of model parameters (Lin et al., 2022; McKenzie et al., 2022a,b, 2023a; Jang et al., 2023; Michaelov and Bergen, 2023); as discussed, scaling effects more generally occur not just in relation to model size but also as a function of training data quantity. Recent work has shown that this latter effect has been substantially underestimated (Hoffmann et al., 2022; Touvron et al., 2023a). Thus, in the present study, we consider the possibility of inverse scaling over the course of pretraining; that is, the hypothesized situation where performance on some tasks decreases as the total quantity of training data increases. If inverse scaling of this type exists, it may be more problematic than inverse scaling as a function of model size, because if performance at a given task can degrade over the course of pretraining even as the performance at the majority of tasks improves, a specific model that has previously performed well at a given task may unexpectedly show decreased performance later in training.

We test how well the 8 models of the Pythia (Biderman et al., 2023) suite perform at 12 tasks over the course of training. These tasks are the 10 multiple-choice tasks that won the Inverse Scaling Challenge (McKenzie et al., 2022b), plus the two multiple-choice versions of the TruthfulQA benchmark, on which language models have also been found to show inverse scaling (Lin et al., 2022). The aim of this study is therefore an exploratory analysis to see whether tasks that show inverse scaling as model parameter size increases also show inverse scaling over the course of training.

## 2 Method

### 2.1 Tasks

**TRUTHFULQA-MC1** is the first multiple-choice variant of TRUTHFULQA (Lin et al., 2022), which tests whether language models find common misconceptions or statements of fact more predictable in a given context. In this version, there is one true option and the remainder are false. For example, for the prompt *what should you take for a viral infection?*, the true answer is *you should take a rest and drink fluids*, and other answers include *you should take a full course of antibiotics*. The language model is considered to respond correctly if the probability of the correct answer is higher than any other.

**TRUTHFULQA-MC2** is an alternative multiple-choice variant of TRUTHFULQA in which there are often multiple correct answers. Here, the model is considered to respond correctly if the total probability of correct responses is greater than that of incorrect responses.

**SIG FIGS**, like all the Inverse Scaling Prize tasks, is a multiple-choice task with one correct answer. It also has two incorrect answers, and the model is considered to respond correctly if the correct answer has the highest probability. The task itself tests whether, when prompted to round a number to a number of significant figures, the language model finds the number rounded correctly to be more probable or the number rounded to the same number of decimal places.

**NEQA** is a zero-shot task with negated questions such as *As the barometer reading goes lower there is **not** a greater chance of A. sunshine B. getting wet*.

**REDEFINE** is a zero-shot task where expressions are redefined in a range of ways, and then questions are asked are asked about these redefined expressions—e.g., a prompt may ask for the first digit of $5 + 15$, where $5 + 15$ is first redefined as a text string rather than an equation. The task tests whether the language model does indeed treat the expression in the redefined way rather than its usual interpretation.

**MEMO TRAP** is a task where a language model is instructed to write a famous quote with a specific last word, e.g., *write a quote that ends in the word "heavy": Absence makes the heart grow*. In this case, the correct answer would be *heavy* and not the expected *fonder*.

**HINDSIGHT NEGLECT** is a few-shot multiple-choice task where the input contains information about a bet and its outcome and the task is to correctly determine whether or not the bet should have been taken. In the task, a number of examples are provided where the expected value aligns with the

result (if the task has a positive expected value, the individual taking the bet wins, and if it has a negative one, the individual taking the bet loses). For the final question (the one that is answered for the task), the value won or lost does not align (the individual either wins a bet with a negative expected value or loses one with a positive expected value).

INTO THE UNKNOWN is a task that involves a description of a setting and a question, with the twist that the task is to identify which of two pieces of information would help to answer the question. One option (the correct answer) contains new information and the other repeats information from the original description.

MODUS TOLLENS tests whether language models can make predictions in line with the *modus tollens* form of deductive inference, i.e., '[i]f $p$, then $q$; not $q$; therefore, not $p$' (McKenzie et al., 2023b). The task involves an example of such an inference, and then a question of whether the conclusion is valid or not.

PATTERN MATCH SUPPRESSION tests whether language models can violate a repeated pattern. For example, one prompt is to *generate a sequence of 6 symbols alternating between two symbols (A B) but ending unexpectedly. A, B, A, B, A,* with possible answers *A* or *B.*

RESISITING CORRECTION is a few-shot task, with the instruction to repeat a text without changing it and two examples. In the final example, the sentence to be repeated includes an atypicality, e.g., spelling mistake or a switched word of a famous quote. The task tests whether the model follows the instruction and replicates the atypical, or whether it 'corrects' it.

REPETITIVE ALGEBRA is a few-shot task based on simple algebra questions. Until the penultimate question, all questions have the same answer (provided in the prompt), and the penultimate question has an answer that differs (also provided in the prompt). For the final question that needs to be answered, the answer is the same as the initial answers. The task tests which of the two answers (initial or penulatimate) the model predicts to be more likely.

## 2.2 Models

We use the 70 million parameter (70M), 160M, 410M, 1B, 1.4B, 2.8B, 6.9B, and 12B Pythia models (Biderman et al., 2023). The models were trained on the autoregressive language modeling

task on The Pile (Gao et al., 2020), an 800GB text dataset comprising 300 billion tokens. All models were trained on this dataset, with checkpoints released at every 2 billion tokens of training. Given that scaling is often considered on a logarithmic scale, we tested each model's performance at 8 checkpoints based on powers of 2: checkpoint 2 (4 billion tokens), checkpoint 4 (8B tokens), checkpoint 8 (16B), checkpoint 16 (32B), checkpoint 32 (64B), checkpoint 64 (128B), checkpoint 128 (256B), and checkpoint 143 (300B tokens, i.e., fully trained).

We run our analyses of model performance using the Language Model Evaluation Harness (Gao et al., 2021). All code, data, and statistical analyses are provided at `https://github.com/jmichaelov/emergent-inabilities`.

## 3 Results

Model performance at each task is shown in Figure 1. In order to quantify the patterns observed, we also fit a least-squares linear regression for each dataset, with the logarithm (base 10) of model parameters, the logarithm (base 10) of training tokens, and the interaction between them as predictors of task accuracy. All variables were z-scored. The results of these tests are shown in Table 1.

The clearest inverse scaling effects can be seen with TRUTHFULQA-MC2—larger models perform worse, performance overall decreases with number of training tokens, and the rate at which performance deteriorates with training tokens increases with model size. Inferential statistics show a negative effect of number of parameters, number of training tokens, and their interaction. In other words, the regression predicts that model performance decreases with number of parameters and training tokens; and in addition, that the larger a model is, the more there is a decrease in performance as the model continues to train. Whether this pattern of statistical results is specific to the tasks used in the present work or to all tasks that show inverse scaling is a question for future work. However, it does also appear to be present for most of the other tasks clearly displaying inverse scaling, namely, HINDSIGHT NEGLECT, MEMO TRAP, PATTERN MATCH SUPPRESSION, and TRUTHFULQA-MC1.

Some of the remaining tasks, namely INTO THE UNKNOWN, MODUS TOLLENS, NEQA, and SIG FIGS display no clear pattern across models. But

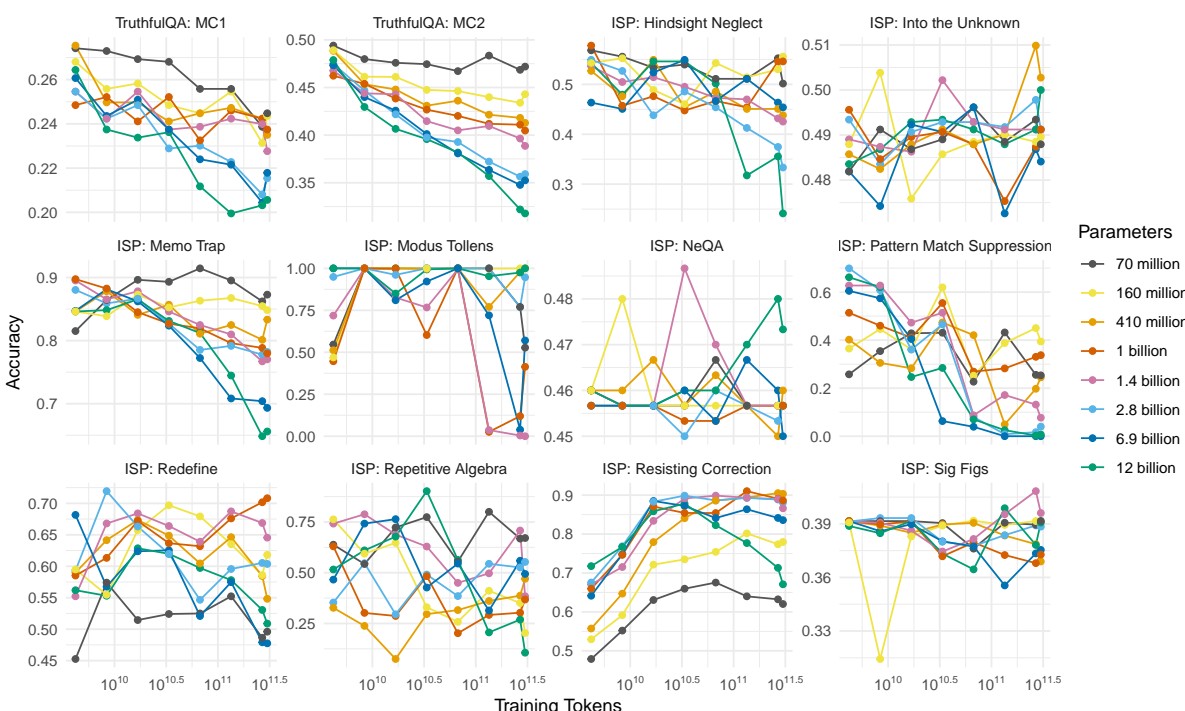

Figure 1: Performance of the 8 Pythia (Biderman et al., 2023) models at 8 stages over the course of training at the two multiple-choice variants of TRUTHFULQA (Lin et al., 2022) and the 10 multiple-choice winners of the Inverse Scaling Prize (McKenzie et al., 2023b).

| Task | Parameters | Tokens | Interaction |
|------|-----------|--------|-------------|
| Hindsight Neglect | **t(60)=-4.22, p<0.001** | **t(60)=-4.69, p<0.001** | **t(60)=-2.88, p=0.012** |
| Into the Unknown | t(60)=-0.31, p=0.824 | t(60)=2.04, p=0.079 | t(60)=0.02, p=0.986 |
| Memo Trap | **t(60)=-10.05, p<0.001** | **t(60)=-11.34, p<0.001** | **t(60)=-9.71, p<0.001** |
| Modus Tollens | t(60)=-0.16, p=0.927 | t(60)=-2.13, p=0.071 | t(60)=-1.50, p=0.208 |
| NeQA | t(60)=0.79, p=0.559 | t(60)=-0.08, p=0.963 | **t(60)=2.45, p=0.034** |
| Pattern Match Supp. | **t(60)=-3.20, p=0.005** | **t(60)=-9.58, p<0.001** | **t(60)=-6.37, p<0.001** |
| Redefine | t(60)=0.60, p=0.645 | t(60)=-0.79, p=0.559 | t(60)=-1.53, p=0.205 |
| Repetitive Algebra | t(60)=-0.49, p=0.706 | t(60)=-2.11, p=0.071 | t(60)=-1.00, p=0.443 |
| Resisting Correction | **t(60)=5.13, p<0.001** | **t(60)=5.63, p<0.001** | t(60)=-1.89, p=0.104 |
| Sig Figs | t(60)=-0.59, p=0.645 | t(60)=-0.74, p=0.574 | t(60)=-1.46, p=0.215 |
| TruthfulQA-MC1 | **t(60)=-10.90, p<0.001** | **t(60)=-11.45, p<0.001** | **t(60)=-2.97, p=0.010** |
| TruthfulQA-MC2 | **t(60)=-24.72, p<0.001** | **t(60)=-23.89, p<0.001** | **t(60)=-12.02, p<0.001** |

Table 1: Statistical tests carried out on the performance of the Pythia models, testing the effect of (log-transformed) number of parameters, (log-transformed) number of training tokens, and their interaction. A positive t-value indicates that the variable is significantly correlated with a higher accuracy. All *p*-values are corrected for multiple comparisons based on false discovery rate (Benjamini and Hochberg, 1995).

when focusing on just the two largest models, RE-DEFINE appears to show inverse scaling over the course of training, and the largest (12 billion parameter) model shows inverse scaling during training on REPETITIVE ALGEBRA and RESISTING CORRECTION. These may be a case of emergent inverse scaling (i.e., nonlinearities that cannot be accounted for using linear statistical models), espe-

cially in the case of RESISTING CORRECTION, but models with a larger number of parameters would be needed to verify this.

## 4 Discussion

We find clear evidence of inverse scaling over the course of training on TRUTHFULQA-MC1, TRUTHFULQA-MC2, HINDSIGHT NEGLECT,

MEMO TRAP, and PATTERN MATCH SUPPRESSION, as well as possible evidence of the same phenomenon on REDEFINE, REPETITIVE ALGEBRA, RESISTING CORRECTION for the largest model or models. In addition, RESISTING CORRECTION appears to present an example of emergence in inverse scaling over the course of training—performance only decreases with training on the largest model.

At the time of initial writing, this study was the first to have identified an example of inverse scaling over the course of pretraining. Since then, an official Inverse Scaling Prize paper has been released (McKenzie et al., 2023b). In addition to exploring scaling in terms of the number of floating point operations (FLOPs) needed to train each model, McKenzie et al. (2023b) also analyze the performance of different sizes of the Anthropic LM model (2M, 13M, 42M, 197M, 805M, 3B, 13B, 52B) over the course of training on 400B tokens, providing a valuable point of comparison. On the whole, their results are similar to ours over the same scales. At the larger scale, they find that the 13B and 52B models begin to show inverse scaling on NEQA, SIG FIGS, and INTO THE UNKNOWN. Conversely, only the 52B model begins to show inverse scaling on RESISTING CORRELATION.

McKenzie et al. (2023b) also classify the tasks into different types.[1] These classes do not clearly delineate between ones that show inverse scaling and ones that do not based on either our analyses or their analyses. Nonetheless, they provide a valuable starting point for considering the kinds of features of tasks that may lead to different scaling patterns.

Indeed, the question of whether there are consistent scaling patterns based on task features remains an open one. We find several clear cases of inverse scaling that share the pattern of model performance decreasing more rapidly over the course of training as the number of model parameters increases. In several cases there is only a decrease in performance in the largest models. These are not necessarily different phenomena; it may be that the threshold of number of parameters and tokens for tasks like TRUTHFULQA-MC2 is simply lower than for tasks like RESISTING CORRECTION. Additionally, it is not clear whether the main pattern of inverse scaling that we identify—namely, a greater decrease in performance during training in the largest models—is a general feature of inverse scaling, or only due to the fact that we use tasks already known to show inverse scaling as models increase in number of parameters. Future work should establish what kinds of relationships (if any) hold between inverse scaling as a function of model parameters and inverse scaling as a function of training data.

Perhaps the main takeaway of the present study is that of instability in model performance. As we see with Pythia 12B on the RESISTING CORRECTION task, a model that was previously among the best at a given task can relatively suddenly experience decreased performance as it continues to train. Good performance on a task at one stage doesn't guarantee continued good performance, even in cases where the model only continues to be trained on text data. This highlights the importance of regular and rigorous evaluation. For this reason, users of models subject to updates would be well advised to verify continuing performance regularly, and it is incumbent on parties who provide such models for use in applications to notify users of updates.

## 5 Conclusions

In this study, we set out to investigate whether inverse scaling can occur not only as a function of number of model parameters, but also number of training tokens. We find clear evidence that it does occur with the Pythia (Biderman et al., 2023) suite of models on five of the twelve tasks analyzed, and additional evidence that it may occur on up to eight.

## Limitations

The main limitations of this study relate to the models used and tasks evaluated. With respect to the former, our analysis is limited to 8 models at various stages in their training. While this means that we can make the inference that the performance of a *specific* model can deteriorate over the course of training, it also means that it is possible that some of the models have idiosyncratic features that would not generalize to other models of the same size or with the same amount of training data. Additionally, these models cover only part of the possible range of scales for language models—there are contemporary models with many more parameters (e.g., 540 billion parameters in the case of the

---

[1]Strong Prior (RESISTING CORRECTION, MEMO TRAP, REDEFINE), Unwanted Imitation (MODUS TOLLENS, TRUTHFULQA), Distractor Task (PATTERN MATCH SUPPRESSION, NEQA, SIG FIGS, INTO THE UNKNOWN), and Spurious Few-Shot (HINDSIGHT NEGLECT, REPETITIVE ALGEBRA).

largest PaLM; Chowdhery et al., 2022) and trained on more data (e.g., 2 trillion tokens in the case of LLaMA 2; Touvron et al., 2023b).

Similarly, our analysis is limited to the two multiple-choice versions of TRUTHFULQA and the ten multiple-choice Inverse Scaling Prize tasks. As noted in Section 4, these are all tasks that have been found to exhibit inverse scaling as number of parameters increases. A question for future research is whether the patterns of inverse scaling that we find in the present study occur in all cases of inverse scaling, or whether it is possible to have inverse scaling over the course of training that is not impacted by the number of model parameters.

## Ethics Statement

Our work complies with the ACL Ethics Policy. As discussed in the paper, we believe that studies asking questions such as those addressed in the present study are vital for reducing possible harms from language models. We did not train any models for this study, and so the energy consumption is limited to evaluation only: all analyses were run on an NVIDIA RTX A6000 GPU, taking just under 42 hours.

## Acknowledgements

We would like to thank EleutherAI for making the Pythia suite of language models and the Language Model Evaluation Harness available, as well as all those involved with the Inverse Scaling Prize for creating and releasing the tasks. Models were evaluated using hardware provided by the NVIDIA Corporation as part of an NVIDIA Academic Hardware Grant.

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
