# OpenReview forum: "Emergent Inabilities? Inverse Scaling Over the Course of Pretraining"
_EMNLP/2023/Conference — EMNLP 2023 Findings_

### Official Review · Reviewer_7HVU · 2023-07-29

**Soundness:** 3

**Excitement:**

2: Mediocre: This paper makes marginal contributions (vs non-contemporaneous work), so I would rather not see it in the conference.

**Paper Topic And Main Contributions:**

Language model performance ($y$) typically improves with increased model size ($m$) and training token amount ($t$) --- a concept known as normal scaling. Recent studies reveal that sometimes, $y$ declines with $m$, particularly in tasks such as negation QA, which is referred to as inverse scaling. This paper reveals $y$ also declines with $t$ dimension, by evaluating the Pythia model family across four tasks that are previously demonstrated inverse scaling over $m$ dimension.

**Reasons To Accept:**

- The finding is interesting and novel. This paper provides another perspective of inverse scaling: performance decreases with training compute, besides previous findings about inverse scaling with model sizes.

- The paper is generally well-written and easy-to-understand.

**Reasons To Reject:**

- The contribution is too limited. The paper only took a pre-trained model family and evaluated them on 4 existing datasets.

- No in-depth analysis. The authors found inverse scaling happens over compute, but why? It would make the paper much more solid if the authors can provide some analysis explaining such training dynamics.



**Reproducibility:**

5: Could easily reproduce the results.

**Reviewer Confidence:**

4: Quite sure. I tried to check the important points carefully. It's unlikely, though conceivable, that I missed something that should affect my ratings.

---

> ### Author Rebuttal · Authors · 2023-08-29
>
> We thank the reviewer for their feedback. Responses to specific comments are given below.
>
> *“The contribution is too limited. The paper only took a pre-trained model family and evaluated them on 4 existing datasets.”*
>
> We appreciate that the reviewer found the finding interesting and novel and agree that the scope of the experiments limits its potential impact. For this reason, we have carried out the same analyses on the 6 remaining multiple-choice Inverse Scaling and TruthfulQA tasks. For the 12-billion-parameter model, we see inverse scaling for both multiple-choice TruthfulQA tasks, as well as the Resisting Correction, Repetitive Algebra, Redefine, Pattern Match Suppression, Memo Trap, and Hindsight Neglect tasks of the Inverse Scaling Prize
>
> As for the concern about using pre-trained models, we find models like these to be frequently used in impactful work to address research questions. In this case, these particular pre-trained models effectively serve the purpose we require of them, which we believe would make training models of a similar scale an unnecessary use of resources.
>
> *“No in-depth analysis. The authors found inverse scaling happens over compute, but why? It would make the paper much more solid if the authors can provide some analysis explaining such training dynamics.”*
>
> We agree that this is indeed a limitation of the original submission and thank the reviewer for pointing it out. As part of the process of remedying this, we have incorporated the feedback of all three reviewers to carry out the same analysis on wider range of tasks, including the remaining 6 multiple-choice Inverse Scaling tasks, TruthfulQA, and other tasks for which inverse scaling has not yet been reported. Based on our previously-mentioned new results, we do see a relatively clear pattern that, of the tasks tested, all the tasks that McKenzie et al. (2023) consider to fall under the “Strong Prior” and “Spurious Few-Shot” type exhibit inverse scaling, but the same is not true of the one “Unwanted Imitation” task (Modus Tollens), and is only true of one of the “Distractor Task” tasks (Pattern Match Suppression). This will be discussed in the revised paper.

---

### Official Review · Reviewer_qpm9 · 2023-08-02

**Soundness:** 4

**Excitement:**

4: Strong: This paper deepens the understanding of some phenomenon or lowers the barriers to an existing research direction.

**Paper Topic And Main Contributions:**

This paper examines the phenomenon of inverse scaling, which is when an LLM's performance on certain tasks decreases as model size grows. This challenges the current observation trend that LLM performance generally increases as model and data size increase.

While previous works have focused on inverse scaling with respect to model size, this paper investigates whether inverse scaling also occurs as training data size increases (i.e. during the course of pretraining itself, when model size is fixed). The authors experiment with the models of the Pythia suite, a collection of 8 LLMs of various sizes with a similar architecture to GPT-3. Each model size of the Pythia suite comes in various checkpoints, each representing a cutoff point after pretraining on a certain amount of tokens. The authors test the models on a set of 4 tasks, which have previously displayed inverse scaling with respect to model size, to investigate whether these tasks also display inverse scaling with respect to data size. They find that 2 of the 4 tasks show inverse scaling with respect to training data size.

The paper contributes with the following findings:

1) Inverse scaling with respect to data size can occur, at least for tasks that already display model size-related inverse scaling.
2) Smaller models display less of this phenomenon than larger models.

**Questions For The Authors:**

A) Why not look at tasks which don't display inverse scaling with respect to model size? Is it possible that these could show inverse scaling with respect to training data size?

**Reasons To Accept:**

The topic of scaling and whether a solved task "stays solved" during the course of pretraining is important, particularly since the current trend is to train models on increasingly large amounts of data. The experiments cover an extensive range of LLM sizes (8 models ranging from 70M to 12B parameters), using an architecture similar to GPT-3, which make this work a valuable foundation for future research. The authors make several important observations, namely 1) some tasks do indeed show inverse scaling as training data size increases, and 2) smaller models have less inverse scaling behavior than larger models. These observations should be investigated further in future works, as they might lead to more insights into LLMs. In particular, it may be the case that some models are "overtrained" and that less pretraining data is in fact more optimal for certain tasks.

**Reasons To Reject:**

Although this is a short paper, the analysis is a bit limited. The authors only present performance on four tasks, and only two of these tasks were found to show inverse scaling with respect to the amount of training data. While this is a promising start, the paper could have provided a broader analysis of the phenomenon that would allow for stronger conclusions. For example, the authors could have experimented on more tasks, particularly emergent tasks (e.g. Wei et al., 2022), to determine the extent of data-related inverse scaling on tasks that LLMs are popularly benchmarked on.

Additionally, it's unclear from the paper whether model size-related inverse scaling occurs independently of data size-related inverse scaling. The paper only focuses on tasks that already display model size-related inverse scaling. But since the data size aspect has not been extensively studied, the paper should either justify why only those tasks were examined, or for the sake of completeness, include tasks that have not yet shown model size-related inverse scaling.

**Reproducibility:**

4: Could mostly reproduce the results, but there may be some variation because of sample variance or minor variations in their interpretation of the protocol or method.

**Reviewer Confidence:**

4: Quite sure. I tried to check the important points carefully. It's unlikely, though conceivable, that I missed something that should affect my ratings.

**Typos Grammar Style And Presentation Improvements:**

Typos:
- Line 33: Duplicate "where"
- Line 58: "specter" should be "spectrum"
- Line 282: "section 1" should be capitalized
- Line 337: "section 4" should be capitalized

Presentation suggestion:
It could be beneficial to do a brief qualitative analysis of the model outputs. This is particularly important for larger models (> 1B), since these can produce acceptable and correct answers from a human perspective, but be automatically evaluated as wrong because they do not adhere to the specific output format of the task. This analysis could have been done for the model checkpoints that showed a significant sudden performance drop, to make sure that the larger models aren't actually doing better than the numerical scores make it seem.

---

> ### Author Rebuttal · Authors · 2023-08-29
>
> We thank the reviewer for their feedback. Responses to specific comments are given below.
>
> *“Although this is a short paper, the analysis is a bit limited. The authors only present performance on four tasks, and only two of these tasks were found to show inverse scaling with respect to the amount of training data. While this is a promising start, the paper could have provided a broader analysis of the phenomenon that would allow for stronger conclusions. For example, the authors could have experimented on more tasks, particularly emergent tasks (e.g. Wei et al., 2022), to determine the extent of data-related inverse scaling on tasks that LLMs are popularly benchmarked on.”*
>
> We agree with the reviewer on this point. The revised version of the paper will include analysis of a wider range of tasks. As noted in the response to anonymous reviewer K1CC, we ran a new analysis on all 10 of the multiple-choice Inverse Scaling tasks and both versions of the multiple-choice TruthfulQA task on a subset of the models and checkpoints (models: 70m, 2.8b and 12b; checkpoints: 4000, 64000, and 143000 (final)) . We found evidence of inverse scaling on the Resisting Correction, Repetitive Algebra, Redefine, Pattern Match Suppression, Memo Trap, and Hindsight Neglect tasks of the Inverse Scaling Prize, and both TruthfulQA tasks. This directly informs the extent of inverse scaling that occurs over training in popular tasks.
>
> *“Additionally, it's unclear from the paper whether model size-related inverse scaling occurs independently of data size-related inverse scaling. The paper only focuses on tasks that already display model size-related inverse scaling. But since the data size aspect has not been extensively studied, the paper should either justify why only those tasks were examined, or for the sake of completeness, include tasks that have not yet shown model size-related inverse scaling.”*
>
> *“A) Why not look at tasks which don't display inverse scaling with respect to model size? Is it possible that these could show inverse scaling with respect to training data size?”*
>
> At the time of writing, the results of the Inverse Scaling Prize challenge only showed evidence of inverse scaling as a function of model parameters. The aim of this study was to test whether there was any evidence at all of inverse scaling over the course of training, and so we selected tasks on which language models had been observed to show inverse scaling of any form as a starting point. In the revised version of the paper, we will discuss the results on the remaining Inverse Scaling tasks and Truthful QA, where, as might be expected given how the tasks were selected, we see the strongest inverse scaling over training for the largest models. We agree with the reviewer that investigating whether the two types of inverse scaling—inverse scaling as a function of number of patterns and inverse scaling as a function of number of training tokens—occur independently would indeed be a valuable direction for future research, and that the emergent tasks reported by Wei et al. (2022) would be a good place to start.
>
> * *'Line 33: Duplicate "where"'*
> * *'Line 58: "specter" should be "spectrum"'*
> * *'Line 282: "section 1" should be capitalized'*
> * *'Line 337: "section 4" should be capitalized'*
>
> We thank their reviewer for noting these.
>
> *“Presentation suggestion: It could be beneficial to do a brief qualitative analysis of the model outputs. This is particularly important for larger models (> 1B), since these can produce acceptable and correct answers from a human perspective, but be automatically evaluated as wrong because they do not adhere to the specific output format of the task. This analysis could have been done for the model checkpoints that showed a significant sudden performance drop, to make sure that the larger models aren't actually doing better than the numerical scores make it seem.”*
>
> We agree that this is a valuable way to holistically evaluate the output of language models, and indeed may reduce under-estimation of their capabilities. However, the tasks investigated in the present study are multiple-choice tasks, and thus the way in which the models are evaluated is not through free generation, but by comparing the relative probabilities of pre-defined continuations or answers. This means that only the probabilities of the correct vs. incorrect generations are taken into account. However, we thank the reviewer for noting this, as it is something important to consider for other forms of evaluation task.

---

### Official Review · Reviewer_K1CC · 2023-08-04

**Soundness:** 3

**Excitement:**

4: Strong: This paper deepens the understanding of some phenomenon or lowers the barriers to an existing research direction.

**Missing References:**

The original Inverse Scaling paper (McKenzie et al., 2013) actually reports model performance over FLOPs, which is a function of the number of training tokens. While the arxiv paper is contemporary, this paper would benefit from discussion of those findings as it is heavily built off this work.

**Paper Topic And Main Contributions:**

There are some tasks known to the NLP community as demonstrating inverse scaling, meaning that performance decreases with greater model scale. These tasks include TruthfulQA and the 11 tasks collected in the Inverse Scaling Prize. However, the community has mainly focused on how performance on these tasks change with model size. This paper instead investigates how performance on 4 of the Inverse Scaling tasks change *over the course of training*. They find that 2 of the 4 tasks demonstrate "inverse scaling."

**Questions For The Authors:**

I found the linear regression experiments very confusing. What is "the interaction between log parameters and log training tokens"? What does t(52) represent (I suspect 52 is the degree of freedom, but where does it come from)? As it is, I was not able to understand the takeaways from this section.

**Reasons To Accept:**

This paper tackles a timely, interesting research question and will spark more interest in the phenomenon of inverse scaling. It may also encourage model developers to publicly release checkpoints (which is not common practice) for the study of training dynamics.

**Reasons To Reject:**

With the limited experimental setup, it is hard to draw any reliable takeaways.
- The authors only study 4 of 11 Inverse Scaling tasks, and do not include other datasets that have demonstrated inverse scaling (e.g., TruthfulQA). With only 4 datasets, it is hard to draw conclusions about training dynamics more broadly, compared to task-specific phenomena.
- The 4 datasets do not demonstrate consistent trends over training, and the authors do not offer discussion of why two tasks in particular show inverse scaling, while the other two do not.
- The largest model studied has 6.9B parameters, while popular LMs these days have reached a scale two orders of magnitude larger. For instance, for some of the Inverse Scaling tasks (like Sig Figs and Into the Unknown), performance does not begin deteriorating until models hit 7B parameters. While such large models are usually not publicly available (and would require significant compute to train from scratch), it does limit the relevance of this paper's study into today's LMs.
- The authors leave many questions open which would make the paper a more complete investigation. For instance, how do trends over training relate to trends over parameter count? There is speculation that different inverse scaling behavior has different underlying causes (e.g, strong priors, or misleading few-shot examples); would these reflect differently in training dynamics?

**Reproducibility:**

5: Could easily reproduce the results.

**Reviewer Confidence:**

4: Quite sure. I tried to check the important points carefully. It's unlikely, though conceivable, that I missed something that should affect my ratings.

**Typos Grammar Style And Presentation Improvements:**

- Line 221: I think this sentence should be the start of the next paragraph.

---

> ### Author Rebuttal · Authors · 2023-08-29
>
> We thank the reviewer for their feedback. Responses to specific comments are given below.
>
> *“With the limited experimental setup, it is hard to draw any reliable takeaways.*
> * *The authors only study 4 of 11 Inverse Scaling tasks, and do not include other datasets that have demonstrated inverse scaling (e.g., TruthfulQA). With only 4 datasets, it is hard to draw conclusions about training dynamics more broadly, compared to task-specific phenomena.*
> * *The largest model studied has 6.9B parameters, while popular LMs these days have reached a scale two orders of magnitude larger. For instance, for some of the Inverse Scaling tasks (like Sig Figs and Into the Unknown), performance does not begin deteriorating until models hit 7B parameters. While such large models are usually not publicly available (and would require significant compute to train from scratch), it does limit the relevance of this paper's study into today's LMs.”*
>
> We thank the reviewer for these comments. We agree that the limited model sizes and number of datasets in the original submission limit the inferences that can be drawn from the results. We are addressing this in two ways. First, have carried out our analyses on the largest Pythia model, which has 12 billion parameters, and thus is closer in size to many of the commonly-used open models today). Second, we have followed the reviewer’s suggestion to include a wider range of tasks.
>
> We have carried out a new analysis along these lines, looking at the 70m, 2.8b and 12b Pythia models at three steps along their learning trajectory: 4000, 64000, and 143000 (final model), and including all 10 multiple-choice Inverse Scaling tasks and both multiple-choice versions of TruthfulQA. Results for the 12-billion-parameter model are clearest—they show inverse scaling over training on the Resisting Correction, Repetitive Algebra, Redefine, Pattern Match Suppression, Memo Trap, and Hindsight Neglect tasks of the Inverse Scaling Prize. We also see inverse scaling over training for both versions of TruthfulQA. We believe that this substantially improves the work: adding a larger model makes the work more relevant to current models; adding more tasks demonstrates the extent of the phenomena and as shown below points to explanations for why inverse scaling over training occurs.
>
> * *"The 4 datasets do not demonstrate consistent trends over training, and the authors do not offer discussion of why two tasks in particular show inverse scaling, while the other two do not.”*
> * *"The authors leave many questions open which would make the paper a more complete investigation. For instance, how do trends over training relate to trends over parameter count? There is speculation that different inverse scaling behavior has different underlying causes (e.g, strong priors, or misleading few-shot examples); would these reflect differently in training dynamics?”*
>
> We hope that our extended analyses can help to shed light on whether there are differences in which kinds of tasks show inverse scaling over parameter count vs. training data. We thank the reviewer for suggesting considering the different types of underlying cause—indeed, our new results described above are striking in that we see inverse scaling for all the tasks we tested that are labeled as falling under the “Strong Prior” and “Spurious Few-Shot” type; but not the “Unwanted Imitation” task (Modus Tollens), and only one of the “Distractor Task” tasks (Pattern Match Suppression). This may be an example of a difference between inverse scaling over parameter count or training tokens. Or it may suggest that there is an as-yet-unidentified feature of the Pattern Match Suppression task leading to inverse scaling. These possibilities will be discussed in the revised version of the paper.
>
> *"I found the linear regression experiments very confusing. What is "the interaction between log parameters and log training tokens"? What does t(52) represent (I suspect 52 is the degree of freedom, but where does it come from)? As it is, I was not able to understand the takeaways from this section.”*
>
> We thank the reviewer for their comment. By the “interaction between log parameters and log training tokens", we mean that in addition to fitting parameters reflecting the relationship between the number of parameters (log-transformed) and accuracy on the one hand and the number of training tokens (log-transformed) and accuracy on the other, the linear regression also fit an additional parameter reflecting the combination of the two, i.e., whether there is a difference in the relationship between the number of training tokens and accuracy depending on the number of parameters.
>
> “t(52)” indicates there are 52 degrees of freedom for the purposes of carrying out the t-test. Degrees of freedom for t-tests are calculated as the number of observations (7 models * 8 checkpoints = 56) - number of predictors (number of parameters + number of training tokens + interaction between the two = 3) - 1 = 52 (see, e.g., https://www3.nd.edu/~rwilliam/stats2/l02.pdf).
>
> *"The original Inverse Scaling paper (McKenzie et al., 2013) actually reports model performance over FLOPs, which is a function of the number of training tokens. While the arxiv paper is contemporary, this paper would benefit from discussion of those findings as it is heavily built off this work.”*
>
> We thank the reviewer for noting this. The McKenzie et al. (2023) paper appears to have been put on arXiv a day before the abstract deadline (and therefore just over a week before the paper deadline), and so we were not aware of this at the time of submission. Our revised draft will engage directly with their findings, which we hope will provide a useful comparison and a starting point for our analysis of model performance performance at tasks they do not investigate (e.g. TruthfulQA).
>
> * *"Line 221: I think this sentence should be the start of the next paragraph.”*
>
> We thank the reviewer for this suggestion and will edit the manuscript accordingly.

---

### Meta-Review · Area_Chair_FAuq · 2023-09-18

**Recommendation:** 3

**Metareview:**

This paper studies inverse scaling behavior over the course of pre-training using a series of Pythia models. They find that for a set of tasks that demonstrate inverse scaling laws (performance decreases as model size increases), models exhibit a decreasing performance over of the course of training. Reviewers have requested more extensive experiments to support the claims made in the paper and authors provide more comprehensive results and discussions in the rebuttal. Thus I recommend acceptance of this paper.

---

### Decision · Program_Chairs · 2023-10-07

**Decision:**

Accept-Findings

**Comment:**

This paper studies inverse scaling behavior over the course of pre-training using a series of Pythia models. They find that for a set of tasks that demonstrate inverse scaling laws (performance decreases as model size increases), models exhibit a decreasing performance over of the course of training. Reviewers have requested more extensive experiments to support the claims made in the paper and authors provide more comprehensive results and discussions in the rebuttal. Thus I recommend acceptance of this paper.